# Study of Annealed Aquivion^®^ Ionomers with the INCA Method [note 1]

**DOI:** 10.3390/membranes9100134

**Published:** 2019-10-17

**Authors:** Stefano Giancola, Raul Andres Becerra Arciniegas, Armand Fahs, Jean-Franҫois Chailan, Maria Luisa Di Vona, Philippe Knauth, Riccardo Narducci

**Affiliations:** 1Institut Català d’investigaciò Química (ICIQ), Av. Països Catalans 16, 43007 Tarragona, Spain; sgiancola@iciq.es; 2University of Rome Tor Vergata, Department of Industrial Engineering and International Associated Laboratory: Ionomer Materials for Energy, Via del Politecnico 1, 00133 Roma, Italy; Raul.becerra@students.uniroma2.eu (R.A.B.A.); divona@uniroma2.it (M.L.D.V.); 3Aix Marseille Université, CNRS, Madirel (UMR 7246), Electrochemistry of Materials Group and International Associated Laboratory: Ionomer Materials for Energy, Campus St Jérôme, 13013 Marseille, France; philippe.knauth@univ-amu.fr; 4Université de Toulon, MAPIEM (EA 4323), CS 60584, 83041 Toulon CEDEX 9, France; Armand.fahs@univ-tln.fr (A.F.); chailan@univ-tln.fr (J.-F.C.)

**Keywords:** proton exchange membranes, PEMFC, PFSA annealing, hydration, n_c_ index

## Abstract

We investigated the possibility to increase the working temperature and endurance of proton exchange membranes for fuel cells and water electrolyzers by thermal annealing of short side chain perfluorosulfonic acid (SSC-PFSA) Aquivion^®^ membranes. The Ionomer n_c_ Analysis (INCA method), based on n_c_/T plots where n_c_ is a counter elastic force index, was applied to SSC-PFSA in order to evaluate ionomer thermo-mechanical properties and to probe the increase of crystallinity during the annealing procedure. The enhanced thermal and mechanical stability of extruded Aquivion^®^ 870 (equivalent weight, EW = 870 g·mol^−1^) was related to an increase of long-range order. Complementary differential scanning calorimetry (DSC) and dynamic mechanical analysis (DMA) measurements confirmed the increase of polymer stiffness by the annealing treatment with an enhancement of the storage modulus over the whole range of temperature. The main thermomechanical relaxation temperature is also enhanced. DSC measurements showed slight base line changes after annealing, attributable to the glass transition and melting of a small amount of crystalline phase. The difference between the glass transition and melting temperatures derived from INCA plots and the ionic-cluster transition temperature derived from DMA measurements is consistent with the different experimental conditions, especially the dry atmosphere in DMA. Finally, the annealing procedure was also successfully applied for the first time to an un-crystallized cast membrane (EW = 830 g·mol^−1^) resulting in a remarkable mechanical and thermal stabilization.

## 1. Introduction

The urgent need for a reduction of pollution and carbon dioxide in the atmosphere together with the price fluctuations of exhaustible fossil fuels, have reinforced the interest in more clean, efficient and sustainable systems for the conversion of energy. Proton exchange membrane fuel cells (PEMFC) and water electrolyzers (PEMWE) [1,2,3] are expected to play a key role in the near future for sustainable energy production and storage.

Despite their huge potential, the high price of produced electricity and hydrogen and durability issues of device components, including the proton exchange membrane, are jeopardizing their wide development. The implementation of highly conductive and durable membranes would thus mark a turning point in the large-scale commercialization of more efficient and long-term technologies. Currently, one of the challenges that researchers still need to face is to combine low membrane resistance with high mechanical strength.

Perfluorosulfonic acids (PFSAs) are the state-of-the-art membranes for PEM technologies due to their outstanding chemical stability, the high proton conductivity in hydrated conditions, the suitable mechanical strength and the low hydrogen and oxygen permeability, used as such and as composites [4,5,6,7,8,9,10,11]. Nafion^®^ long side chain PFSA (LSC-PFSA) has been the most investigated ionomer for both PEMFC and PEMWE [12,13,14,15,16,17].

In recent years, ever-increasing attention has been directed towards perfluorinated ionomers with shorter and non-branched pendant side-chain (SSC-PFSA; e.g., Aquivion^®^) [18,19,20,21]. The higher crystallinity of SSC-PFSA (e.g., Aquivion^®^ EW = 870) [22] with respect to LSC-PFSA [23] allows it to shift the balance between proton conductivity and mechanical properties at lower equivalent weight (EW). This ultimately results in better transport properties and higher operating temperature of PFSA membranes. Figure 1 shows the structures of Nafion^®^ EW 1100 and Aquivion^®^ EW 830.

Despite the tremendous research effort, PFSA properties are not yet fully understood [23]. A deep PFSA comprehension is still needed to exceed the current membrane performance. In this context, we decided to start a systematic research on Nafion^®^ chemical-physical fundamental properties (e.g., memory at room temperature of the water-uptake at higher temperatures, effect of the equivalent weight on the ionomer melting point (*T*_m_), precise relation between equilibrium relative humidity (RH) and water-uptake at the various temperatures of operation, effect of thermal treatments and so on).

We introduced a new methodology, called INCA (Ionomer n_c_ Analysis), where n_c_ is an index proportional to the force of the ionomer matrix, which balances the osmotic pressure of the inner proton solution at the different temperatures. The determination of the hydration number and the volume of the samples allow us to find the density, the molality and molarity of the internal solution in addition to the internal osmotic pressure [24,25,26,27,28,29,30,31,32]. The INCA method is based on the use of n_c_/T plots to determine some fundamental properties of an ionomer, such as the degree of crystallinity and the melting temperature. n_c_ is directly proportional to Young′s modulus: one n_c_ unit is equivalent to an increase of Young′s modulus by about 6.5 MPa. n_c_ depends on both RH and temperature [25]. In particular, it decreases by increasing both T and RH. It is thus a mechanical index characterizing the membrane mechanical properties at fixed hygrothermal conditions. A high n_c_ means good mechanical properties, related to high performances.

Today′s tendency is to maximize the efficiency of PEM devices by reducing both membrane thickness and EW. On the one hand, thin membranes based on low EW ionomers offer advantages in terms of reduced electrical resistance and enhanced hydration especially at low relative humidity (RH). On the other hand, they suffer from poor mechanical strength and high gas crossover. Membrane mechanical properties are thus currently crucial and a consequent part of the effort of researchers is focused on PFSA mechanical stabilization [33].

A further aspect to consider is the morphological stability of PFSA. Alberti et al. reported that when the relative humidity and temperature (RH/T couples) exceeds certain critical values, irreversible processes could take place and provoke Nafion^®^ morphological change leading to a severe reduction of the membrane through-plane conductivity [28,34]. This aspect is crucial especially in PEMWE where the membrane is exposed to both high relative humidity (100% RH) and temperatures (T ≥ 80 °C) and thus subject to very high compressive stress [35].

Thermal annealing is a procedure used to obtain PEM mechanical stabilization [36,37] and reduced gas permeability [38]. This powerful physical reinforcing strategy can be easily applied to both preformed and cast membranes. Moreover, it can be used also in combination with other reinforcing methodologies [18,21,39]. However, annealing has been usually performed in an empirical way through a short treatment at high temperatures (160–200 °C). Some years ago, due to the use of the INCA method, we started a thorough investigation of the annealing of extruded Nafion^®^ 1100 driven by the desire to rationalize it and maximize its efficiency [27,31]. We assumed that the mechanical stabilization was induced by an increased crystallization of a pre-existing semi-crystalline phase due to a macromolecular rearrangement. The annealing temperature (*T*_an_) must be chosen between the ionic-cluster transition temperature (*T*_α_) and the melting temperature (*T*_m_) of the crystalline phase. Dimethyl sulfoxide (DMSO), which has a high boiling temperature, was added as proton acceptor and plasticizing agent, with the aim to promote the solid-state macromolecular rearrangement. The annealing treatment resulted in a significant enhancement of Nafion^®^ mechanical strength [31,40]. Moreover, no loss of through-plane proton conductivity was observed despite the lower water content of the annealed sample [31]. This might be ascribed to the reduced tortuosity of the material and/or the reduction of hydrophilic domain dimensions translating respectively in improved proton mobility and/or concentration. Based on the INCA analysis, we were able to select and perform a Taylor-made thermal treatment. Very interestingly, annealing allowed to sensibly reduce the formation of low conductive ribbon type morphologies occurring in membranes constrained to swell between two plates at 100% RH and 100 °C. This morphological stabilization prevented the drastic through-plane conductivity drop observed in non-annealed films due to the ribbon formation/reorganization parallel to the membrane surface [28,41]. We would also like to point out that membranes that do not have nanometric or micrometric phases dispersed within them (composite) cost less and can also be more easily reused.

In this paper we apply, for the first time, our INCA analysis to both extruded and cast Aquivion^®^ membranes of different EW with the aim to evaluate their fundamental properties and extent our method also to SSC-PFSA. We are thus able to design a proper annealing treatment to stabilize mechanical properties of both preformed and cast Aquivion^®^ membranes.

## 2. Experimental

### 2.1. Chemicals

Aquivion^®^ water dispersions D79-25BS (EW = 790 g·mol^−1^), D83-24B (EW = 830 g·mol^−1^), Aquivion^®^ extruded membranes E87-12S (EW = 870 g·mol^−1^, 120 μm thick), E98-05 (EW = 980 g·mol^−1^, 50 μm thick) and other reagents were supplied by Sigma-Aldrich (St. Louis, MO, USA).

### 2.2. Membrane Preparation

#### 2.2.1. Extruded Semi-Crystalline Aquivion^®^ 870 and 980

Before testing, Aquivion^®^ 870 and 980 membranes were first treated for 2 h in 1 M sulfuric acid at room temperature (RT) and then washed in deionized water (DW) several times for 24 h.

#### 2.2.2. Un-Crystallized Aquivion^®^ 790 and 830

Aquivion^®^ dispersions (12 wt %) in a mixture of 1-propanol/water (70/30 wt %) were casted with a doctor blade; the solvent was evaporated first in air for 24 h at room temperature and then for 1 h at 80 °C in a ventilated oven. Before testing, Aquivion^®^ 790 and 830 cast membranes were treated for 2 h in 1 M sulfuric acid at RT and then washed in DW several times for 24 h.

#### 2.2.3. Annealed Semi-Crystalline Aquivion^®^ 870 and 980

Annealed Aquivion^®^ 870 was prepared as follows [27]: a large batch of 1 M solution of DMSO in ethanol was prepared. One piece of anhydrous Aquivion^®^ 870 was cut and weighed corresponding to 0.5 meq (i.e., 0.435 g). This membrane was then placed inside a Teflon bottle. One mL of 1 M ethanol solution of DMSO was added in the vessel to give a calculated value of λ(DMSO) = 2.0 ± 0.2 where λ(DMSO) are moles of DMSO per EW of ionomer. The vessel was closed and the solution was left to equilibrate with the ionomer membrane for about 1 h at room temperature. After evaporation of the ethanol solution under moderate agitation at 80 °C, the vessel was closed again and placed in an oven at 140 °C for the desired time. After cooling, the membrane was treated for 2 h in 1 M sulfuric acid at RT and then washed in DW several times for 24 h. For Aquivion^®^ 980, the same procedure was followed as for Aquivion^®^ 870 except that the oven treatment was performed at 150 °C for 7 days.

#### 2.2.4. Annealed Aquivion^®^ 830

Cast Aquivion^®^ 830 was first treated at 135 °C for 15 h and then left to equilibrate for 1 h at RT in a 1 M solution of DMSO in water λ(DMSO) = 2.0 ± 0.2, we used an aqueous solution because in the ethanol solution the swelling was excessive). After evaporation of the water solution under moderate agitation at 80 °C, the membrane was annealed in a closed vessel at 135 °C for the desired time. After cooling, the membrane was treated for 2 h in 1 M sulfuric acid at room temperature (RT) and then washed in deionized water several times for 24 h at RT.

### 2.3. Characterization

#### 2.3.1. Water Uptake and n_c_ Measurements

The previous materials were treated in liquid water for 600 h at different temperatures inside a Teflon container to determine the n_c_ index.

After this equilibration, the membranes were kept at 25 °C for 24 h in a closed Teflon vessel [24]. The excess of water was carefully wiped off with filter paper and the membrane mass was determined (*m_wet_*) using a weighing bottle and an analytical balance; then the samples were dried over P_2_O_5_ for 3 days and weighed (*m_dry_*):(1)$$WU=\frac{m_{wet}-m_{dry}}{m_{dry}}\times 100$$

The hydration number was calculated as:(2)$$\lambda =\frac{n\left(H_{2}O\right)}{n\left(SO_{3}H\right)}=\frac{WU}{IEC\times M\left(H_{2}O\right)}\times 1000$$

The uncertainty is about 0.5.

The λ values were converted into n_c_ values by the Equation (3):(3)$$n_{c}=\frac{100}{\lambda -6}$$

This equation is valid for λ ≥ 10 as derived in references [24,25].

#### 2.3.2. Differential Scanning Calorimetry (DSC)

Differential scanning calorimetry (DSC) was performed on a DSC Q100 apparatus (TA Instruments, New Castle, DE, USA). The scans were carried out under a nitrogen purge, and pristine and annealed Aquivion 870 samples (6 mg) were placed in holed aluminum pans. An empty pan was used as the reference. The samples were heated from 20 to 200 °C with a scanning rate of 10 °C/min. To avoid relaxation effects, the glass transition temperature was determined on the second cycle, so that the free water or solvent is evaporated. The midpoint temperature of the heat flow jump was taken as the glass transition temperature (*T*_g_).

#### 2.3.3. Dynamic Mechanical Analysis (DMA)

Dynamic mechanical analysis (DMA) was performed on a DMA Q800 apparatus (TA Instruments, New Castle, Delaware, United States) in extension mode with samples of approximately 12 mm × 7 mm size and 120 μm thickness. The DMA was operated in air at a fixed frequency of 1 Hz with 0.5 N initial static force, force track 125% and an oscillation amplitude of 10 μm. This last value was chosen to keep the linear viscoelastic domain of samples during the experiments. The measurements were conducted with 3 K/min heating rate between 20 and 200 °C [31]. The relaxation temperature (*T*_α_) was considered as the maximum of tan δ.

## 3. Results and Discussion

### 3.1. n_c_/T Plots of Un-Annealed and Annealed Semi-Crystalline Extruded Aquivion^®^ 870 and 980

Extruded PFSA membranes generally had higher mechanical strength and ductility compared to cast ones of same EW. Due to these outstanding properties, extrusion still remained the state-of-the-art fabrication technique for perfluorinated ionomer membranes. For this reason, directly enhancing the mechanical properties of preformed extruded films would be enormously beneficial. We decided thus to investigate the annealing behavior of Aquivion^®^ 870, the commercial membrane with the lowest EW and the heat of the fusion similar to that of Nafion^®^ 1100.

Figure 2 shows the n_c_/T plots in liquid water of Aquivion^®^ 870 after annealing at 140 °C in presence of DMSO for respectively 3 and 7 days.

n_c_/T plots are characterized by single or multiple straight lines. The extrapolated temperature on the T axis (n_c_ = 0) of each line can be linked to a characteristic ionomer temperature at certain hygrothermal conditions. We remember from the thermodynamic and statistic theory of elastomer deformation that for a semi-crystalline polymer Young′s modulus decreased linearly to zero at the melting temperature [42].

The n_c_ of the un-annealed Aquivion^®^ linearly decreased from 50 to 120 °C and the extrapolation of the plot to the T axis (n_c_ = 0) results to be 155 °C. This temperature was attributed to the melting temperature of an ionomer crystalline phase already present in the as-received membrane [27]. Crystallinity has been indeed detected also in preformed Aquivion^®^ 870 membranes like in other extruded PFSA films [43]. Based on this assumption, we selected 140 °C as the proper annealing temperature, closer to *T*_m_; for Nafion^®^ 1100 the temperature of 130 °C was also used, obtaining for the same time of treatment a lower stabilization [27]. This temperature is indeed slightly lower than the ionomer *T*_m_. Encouraged also by the success already obtained with Nafion^®^ and confirmed by DSC analysis [27], we decided to perform the thermal treatment in presence of DMSO in order to facilitate the macromolecular rearrangement and promote a better crystallization. As shown in Figure 2, the extrapolation of the plots of un-annealed and annealed samples converge at the same temperature (155 °C). Moreover, the annealing treatments remarkably shift the plots towards the right side and increase the absolute slopes.

This indicates a noteworthy enhancement of membrane mechanical strength, also at high temperature, ascribed to an increase of the sample crystallinity. The shift towards right increased by increasing the time of the treatment from 3 to 7 days. It is also possible to quantify the annealing treatment by taking in consideration the change of slope of the plots. In particular, after annealing at 140 °C for 7 days, an increase of slope by 240% was observed.

Using the relation of the osmotic pressure π of a solution with its concentration c, where R is the ideal gas constant and T the absolute temperature:(4)π=cRT
one can easily derive Equation (5), given the proportionality of the n_c_ index with the osmotic pressure [29]:(5)$$\frac{\Delta n_{c}}{\Delta T}=\mathrm{kcR}$$
where k is the proportionality constant between n_c_ and the osmotic pressure and c is the concentration of the inner proton solution. The absolute slope increases during annealing could thus be ascribed to an increase of the inner proton concentration due to the reduction of the volume of cluster domains. However, we were presently unable to calculate exactly the proton concentration from simple n_c_/T plots because of the lack of precise morphological information [29] of PFSA ionomer like Nafion^®^ and Aquivion^®^.

Alberti et al. for Nafion^®^ 1100 have reported similar annealing behavior. Moreover, both Aquivion^®^ 870 and Nafion^®^ 1100 have similar extrapolated melting temperatures indicating similar crystallite morphology. This can be ascribed to the similar backbone length (polytetrafluoroethylene PTFE repeat unit) of the two ionomers [43].

Figure 3 shows the n_c_/T plots in liquid water of thin Aquivion^®^ 980 membranes, before and after annealing at 150 °C in presence of DMSO for 7 days. Similarly to Aquivion^®^ 870, the n_c_ of the un-annealed membrane decreased linearly in the 50–130 °C temperature range with a plot extrapolation (n_c_ = 0) of 160 °C. Also in this case, this temperature was attributed to the melting temperature of a preformed ionomer crystalline phase [27]. The extrapolation of the plots of both un-annealed and annealed membranes converged at the same temperature (160 °C). The treatments resulted in an enhancement of the membrane mechanical strength proved by a plots shift towards the right side due to the increase of the absolute slopes. The melting temperature was slightly higher than that of Aquivion^®^ 870 due to a higher EW, but the polymer had higher water content throughout the temperature range, evidently due to a lower initial crystallinity.

### 3.2. n_c_/T Plots of Un-Crystallized Aquivion^®^

n_c_/T plots in liquid water of un-crystallized Aquivion^®^ 790 and 830 were evaluated and compared with that of un-crystallized Nafion^®^ 1100 already reported [29] (Figure 4).

For SSC-PFSA, a linear plot was observed by increasing the temperature from 40 to 80 °C with extrapolated temperatures around 105 °C for Aquivion^®^ 790 and 830. This temperature was similar to that of un-crystallized Nafion^®^ 1100 (105–110 °C). Very interestingly, the extrapolated temperatures of the n_c_/T plots were similar to the ionic-cluster transition temperature (*T*_α_) obtained by dynamic mechanical analysis (DMA) for low hydrated membranes [31,43]. This temperature has been associated to the onset of long range chain mobility occurring as a result of destruction of electrostatic interactions, including hydrogen bonds and van der Waals interactions present between the chains [23]. Although a direct comparison between INCA and DMA methods is not yet possible due to the different membrane water content and the lack of experimental results on completely amorphous films, we supposed that the extrapolated temperature of the n_c_/T plot would coincide with *T*_α_ at certain membrane hydration conditions. *T*_α_ has been reported to decrease with increasing membrane hydration due to the shielding effect of water acting as plasticizer [23].

The INCA method results thus to be a powerful analytical tool also able to determine ion-cluster transition temperatures (*T*_α_) of both semi-crystalline and un-crystalline PFSA Nafion^®^ and Aquivion^®^ membranes. Moukheiber^43^ reported for semi-crystalline Aquivion^®^ a *T*_α_ decreasing with decreasing EW. He attributed this behavior to the reduced ionomer crystallinity promoting chain motion.

### 3.3. n_c_/T Plots of Un-Crystallized and Annealed Aquivion^®^ 830

Figure 5 displays the evolution of n_c_/T plots of cast un-crystallized Aquivion^®^ 830 after treatment at 135 °C for 15 h and annealing at 135 °C for 7 days in presence of DMSO. We selected for Aquivion^®^ 830 an annealing temperature slightly lower than that used for Aquivion^®^ 870 assuming also a lower melting temperature for the former due to its higher branching degree hindering chain packing and crystallization and due to the lower EW. This behavior is widely common in branched polymers [42]. The dependence of the melting temperature on the EW was verified by Alberti et al. for extruded Nafion^®^ 1100 and 1000 with a decrease of 10 °C for the lowest equivalent weight. The same was checked for Aquivion extruded membranes with different EW: they presented different *T*_m_ and as the EW increased the *T*_m_ increased accordingly. This knowledge is useful for choosing the annealing temperatures.

As depicted in Figure 5, the n_c_/T plot of the un-crystallized Aquivion^®^ shift towards right after treating at 135 °C for 15 h with no change of the extrapolated ionic-cluster transition temperature. Based on the completely amorphous character of the as-cast film, we ascribed the membrane mechanical stabilization to morphological PFSA changes with no crystallinity onset. It is widely accepted that PFSA ionomers are in a quasi-equilibrium state with long relaxation time [23]. Thermal treatment could thus promote the change to a more entangled morphological state due to macromolecular motion or/and decrease the internal volume. A remarkable improvement of PFSA mechanical properties was observed after annealing in presence of DMSO resulting in a significant right shift of the plot. Moreover, the extrapolated *T*_α_ also increased to 125 °C.

### 3.4. Dynamic Mechanical Analysis (DMA) of Pristine and Annealed Aquivion^®^ 870

The DMA analysis allows distinguishing the elastic response (storage modulus E’) and the viscous response (loss modulus E”) of the polymers. In the solid state, the elastic part is much higher than the viscous part; the storage modulus is close to the Young modulus obtained from static tensile tests. The ratio between the loss modulus and the storage modulus is the damping (tan δ), a good parameter to find the relaxation phenomena. The largest peak, denominated α, is assigned to the main relaxation process, which is associated to the glass transition in most polymers and concerns the global amorphous phase. In the case of ionomers (Nafion, etc.), this phenomenon is largely impacted by ionic regions, which are sensitive to the water content, the degree of neutralization and/or the ion type. This relaxation is due to main and side chain motions within or near the ion-rich domains [44].

Figure 6 and Table 1 present the DMA results for Aquivion^®^ 870 before and after annealing at 140 °C in the presence of DMSO for 3 days. The main tan δ peak attributed to the α relaxation phenomenon was around 124 °C for pristine Aquivion^®^ 870 and increased to 131 °C for the annealed sample. The annealing evidently enhanced the long-range order in the ionomer, which hindered chain motions, which need more energy to move, leading to a higher relaxation temperature. One can note also that the storage modulus was higher for the annealed sample than for the pristine one over the whole range of temperature from the glassy state to the rubbery state. The slight decrease of the α peak intensity after the thermal treatment was consistent with an increase of the stiffness of the polymer, due to the better long-range order after annealing. Finally, the sharp decrease of the storage modulus at the end of the experiment was attributable to the melting of the polymer at around 185 °C.

The main limitation of the DMA technique, used essentially for non-ionic polymers, is that the measurements are performed in dry conditions, which levels the effect on the main relaxation process. The higher melting temperature observed in DMA vs. INCA plots might be attributed to the absence of water, which increased the interactions between polymer chains and also to the dynamic conditions in DMA related to the heating rate. The INCA method reflected the mechanical properties in equilibrium in liquid water at working temperature and could be a complementary technique to a DMA analysis in order to study different ionomer samples.

### 3.5. Differential Scanning Calorimetry (DSC) of Pristine and Annealed Aquivion^®^ 870

Figure 7 shows the DSC curves obtained for pristine and annealed material. The first run (presented only for the pristine sample) shows the endothermic peak due to the evaporation of water below 140 °C. In the second run, this peak was absent and one could observe slight changes of slope in the annealed sample, at around 120 and 170 °C, which could be attributed, respectively, to the glass transition of the ionomer amorphous phase and the melting temperature of a small amount of crystalline material. These endothermic transitions were related to a higher long range order (consistent with the increase of *T*_α_ in DMA) and a slightly increased crystallinity (in accordance with *T*_m_ in DMA) after treatment with DMSO, a behavior similar to that seen previously in Nafion 1100 [27]. The crystallinity formed during annealing corresponds to only small portions of the ionomer chains and a large part of amorphous ionomer remains, therefore, linked to the crystalline part. Since crystalline components are joined between them by the amorphous portions, the chains separation becomes only possible after the melting of the crystalline component [27].

## 4. Conclusions

In this work, the INCA (Ionomer n_c_ Analysis) method, first developed for Nafion^®^, was extended to Aquivion^®^ short side chain perfluorosulfonic acid membranes with the aim of understanding their thermo-mechanical properties in specific hygrothermal conditions, and properly improved their mechanical stability by thermal annealing with a plasticizing solvent. Semi-crystalline Aquivion^®^ 870 (EW = 870 g·mol^−1^) shows a linear n_c_/T plot in the examined temperature range (50–120 °C) with an extrapolated temperature of 155 °C (n_c_ = 0) corresponding to its melting point. After annealing at 140 °C in presence of DMSO as annealing agent, we observed an outstanding thermo-mechanical membrane stabilization represented by a remarkable slope increase of the n_c_/T plot. Similar behavior was verified for Aquivion^®^ 980. The INCA analysis of un-crystallized low-temperature casted Aquivion^®^ 830 (EW = 830 g·mol^−1^) and Aquivion^®^ 790 (EW = 790 g·mol^−1^) showed linear plots with extrapolated temperature of 105 °C. Based on the completely amorphous membrane structure, we associated this temperature to the ionomer transition temperature *T*_α,_ largely impacted by ionic regions. Similar temperatures were observed for un-crystallized Nafion 1100 (≈110 °C). DMA and DSC experiments showed *T*_α_ values, which were slightly higher, due to the dry conditions, which enhanced the interactions between chains that need more energy to move. The melting temperature was also enhanced. INCA is thus a powerful analytical tool to evaluate ionic-cluster transition temperatures of amorphous PFSA and *T*_m_ for semi-crystalline materials at a certain relative humidity. Finally, an annealing treatment in the presence of DMSO was successfully applied also to cast Aquivion^®^ 830. These results corroborated the reliability and versatility of this method to enhance the mechanical properties of both extruded and cast PFSA membranes of different EW. From the point of view of mechanical properties, the best membrane was the annealed Aquivion^®^ 870, however depending on the temperature of use other materials and treatments may be taken into consideration.

A further improvement of SSC PFSA membranes is possible by optimization of the annealing conditions in presence of an appropriate solvent.

## Figures and Tables

**Figure 1 membranes-09-00134-f001:**
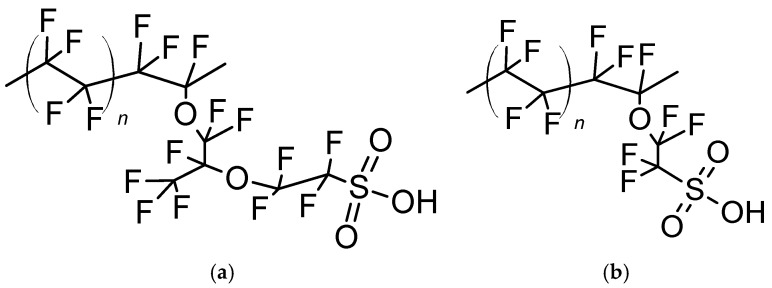
(**a**) Long side chain perfluorosulfonic acid (LSC-PFSA) Nafion^®^ 1100 (*n* = 6.6) and (**b**) short side chain PFSA (SSC-PFSA) Aquivion^®^ 830 (*n* = 5.5).

**Figure 2 membranes-09-00134-f002:**
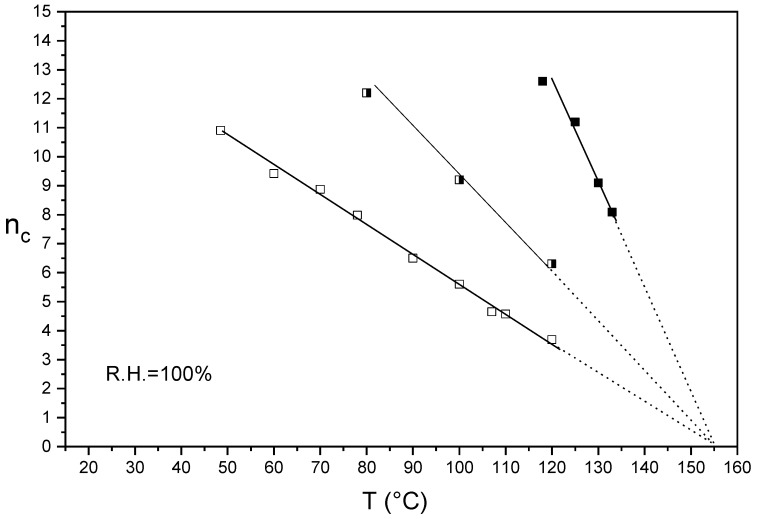
n_c_/T plot of semi-crystalline Aquivion^®^ 870 before (**empty squares**) and after annealing at 140 °C in presence of DMSO for 3 (**half empty squares**) and 7 days (**full squares**).

**Figure 3 membranes-09-00134-f003:**
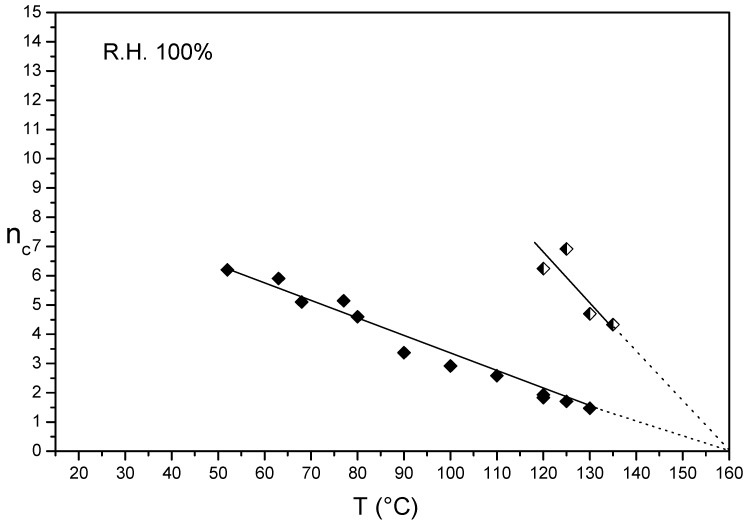
n_c_/T plot of semi-crystalline Aquivion^®^ 980 before (**full rhombus**) and after annealing at 150 °C in presence of DMSO for 7 days (**half empty rhombus**).

**Figure 4 membranes-09-00134-f004:**
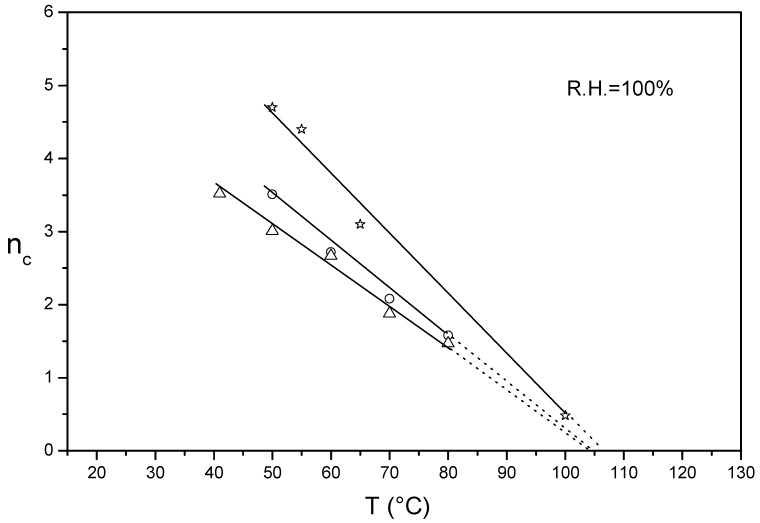
n_c_/T plot of un-crystallized Aquivion^®^ 790 (**empty triangles**), Aquivion^®^ 830 (**empty circles**) and Nafion^®^ 1100 (**empty stars**).

**Figure 5 membranes-09-00134-f005:**
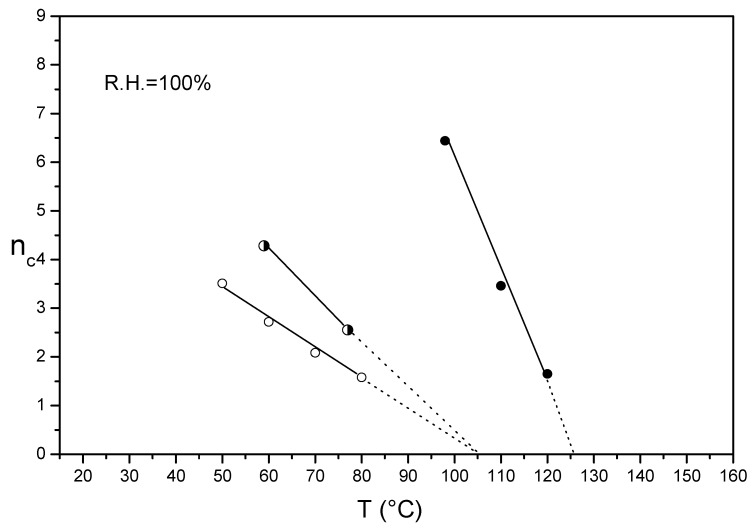
n_c_/T plot of un-crystallized Aquivion^®^ 830: as cast (**empty circles**), after treatment at 135 °C for 15 h (**half empty circles**) and treated at 135 °C for 15h and annealed at 135 °C in presence of DMSO for 7 days (**full circles**).

**Figure 6 membranes-09-00134-f006:**
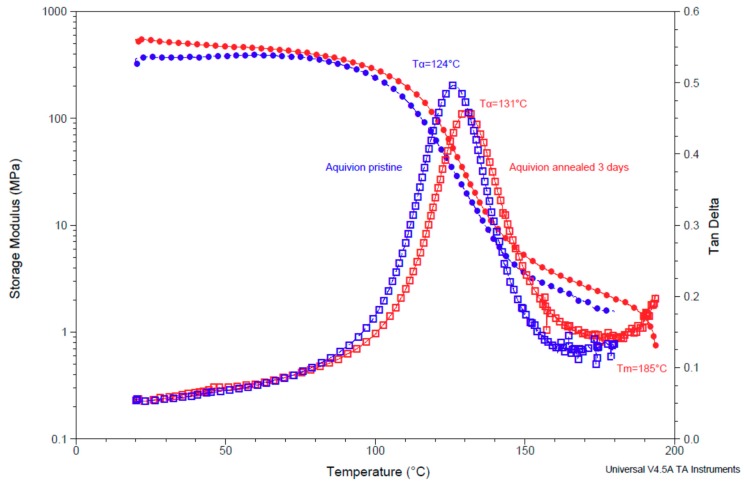
Dynamic mechanical analysis (DMA) curves of Aquivion^®^ 870 before (**blue**) and after annealing at 140 °C in presence of DMSO for 3 days (**red**).

**Figure 7 membranes-09-00134-f007:**
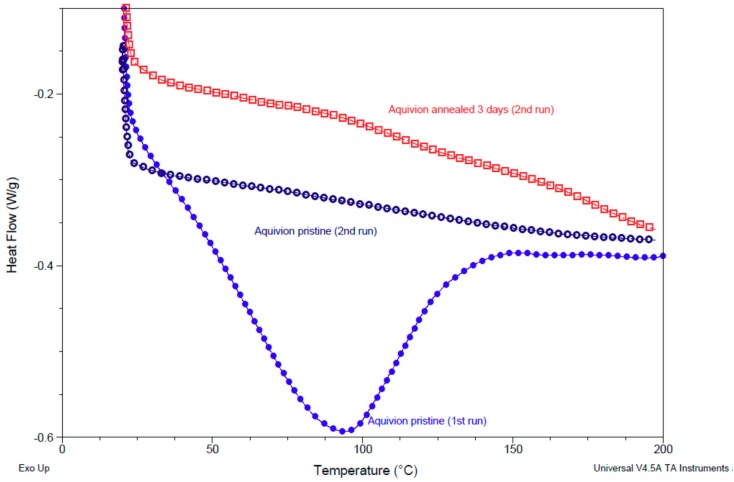
Differential scanning calorimetry (DSC) curves of Aquivion^®^ 870 before (**blue**) and after annealing at 140 °C in presence of DMSO for 3 days (**red**).

**Table 1 membranes-09-00134-t001:** Ionomer relaxation temperatures *T*_α_ (°C), maximum damping intensity I (tan δ) and storage modulus E’ of Aquivion^®^ 870 before and after annealing at 140 °C in presence of DMSO for 3 days.

Sample	*T*_α_/°C	I (tan δ)	E′/MPa (25 °C)	E’/MPa (50 °C)
Aquivion 870 pristine	124 ± 3	0.49	350 ± 30	360 ± 40
Aquivion 870 3 days annealed	131 ± 1	0.46	540 ± 10	490 ± 30

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
