# Peer review of "Study of Annealed Aquivion® Ionomers with the INCA Method"

_membranes, 2019, doi:10.3390/membranes9100134_

Round 1

Reviewer 1 Report

After reviewing the manuscript, I regret that it is not acceptable for publication in its current form. The topic of study is very relevant to the journal and the application in PEM fuel cells. However, this study is lack of novelty and did not address important issues which I’ll elaborate below.

The authors cited the work of Alberti et al. where it was made clear that the INCA method would need other methods, such as DSC and DMA, to characterize polymers. In this manuscript, conclusions or claims by the authors need to be supported with alternative methods to improve the credibility. For example, the measurement of glass transition temperature, melting temperature, and ionic-cluster transition temperature can be obtained from DSC or DMA. It is questionable or unclear regarding why the slope of nc/T matters. The definition of nc does not involve temperature. More elaboration/derivation is needed to explain how the nc/T slope is translated to mechanical properties of the polymer. One important aspect of membrane for fuel cell application is gas permeability. How does annealing affect gas permeability for the membrane? It is not obvious to the reader how Nafion® membrane behaves differently with Aquivion® during the annealing process. The role of side chain length is an important aspect to be addressed here.

Reviewer 2 Report

Minor points:

Line 20-21: “…the increase of the original semi-20 crystalline phase.” It is confusing about the word “original” since the post cast heat treatment is not related to original. Line 21: “mechanical properties” must be specified. Line 62-63: An additional explanation about “the inner proton solution” is recommended. Line 90-91: Regarding the concept of “the ionic cluster transition temperature”, any previous publications have established this concept? If not, what is its meaning in polymer physics? Line 97, mechanistically, it is hard to establish the relation for why annealing treatment causes reduction of hydrophilic domain dimensions, and how does the later favour proton conduction.

Major points:

Line 100, why were the “low conductive ribbon type morphologies” not further elaborated in the text? Experimental section must include details of how Nc is measured. The manuscript claims the designed annealing brings about an increase in crystallinity in the designated PEM, authors have to provide a direction experimental evidence, such as DSC or X-ray diffraction or SAXS, to support it. There is an obvious lack of a clear description about the role of DMSO in the heat treatment process.

Reviewer 3 Report

This manuscript is well addressed for the mechanical properies of SSC PFSAs in terms of anneling predicted by INCA. The extensively useful ideas for the improvement of mechanical stability as well as the prevention of gas permeability by means of annealing at proper temperatures for various SSC PFSA especially Aquivion series. Before publication some minor things should be improved for better readership.

1. The decision of annealing temperature on the basis of INCA results was a little explained but more detail strategy for the selection of optimal annealing temperatures should be given for each annealing temperatures for un crystallized and crystallyzed Aquivions.  

2. Please explain much more details on why the authors selected DMSO as a platicizing solvent. In addition other platicizing solvents should be also given.

Reviewer 4 Report

The text merit publication after an extensive and careful revision of the flaws indicated in the annotated file - please see the attached file.

Round 2

Reviewer 1 Report

I thank the authors for considering my comments and performing make-up tests. 

Reviewer 2 Report

The experimental section has been revised to provide the procedure about annealing the two samples.